# Can $Q$-Learning with Graph Networks Learn a Generalizable Branching Heuristic for a SAT Solver?

**Vitaly Kurin**[*]
Department of Computer Science
University of Oxford
Oxford, United Kingdom
vitaly.kurin@cs.ox.ac.uk

**Saad Godil**
NVIDIA
Santa Clara, California
United States
sgodil@nvidia.com

**Shimon Whiteson**
Department of Computer Science
University of Oxford
Oxford, United Kingdom
shimon.whiteson@cs.ox.ac.uk

**Bryan Catanzaro**
NVIDIA
Santa Clara, California
United States
bcatanzaro@nvidia.com

## Abstract

We present Graph-$Q$-SAT, a branching heuristic for a Boolean SAT solver trained with value-based reinforcement learning (RL) using Graph Neural Networks for function approximation. Solvers using Graph-$Q$-SAT are complete SAT solvers that either provide a satisfying assignment or proof of unsatisfiability, which is required for many SAT applications. The branching heuristics commonly used in SAT solvers make poor decisions during their warm-up period, whereas Graph-$Q$-SAT is trained to examine the structure of the particular problem instance to make better decisions early in the search. Training Graph-$Q$-SAT is data efficient and does not require elaborate dataset preparation or feature engineering. We train Graph-$Q$-SAT using RL interfacing with MiniSat solver and show that Graph-$Q$-SAT can reduce the number of iterations required to solve SAT problems by 2-3X. Furthermore, it generalizes to unsatisfiable SAT instances, as well as to problems with 5X more variables than it was trained on. We show that for larger problems, reductions in the number of iterations lead to wall clock time reductions, the ultimate goal when designing heuristics. We also show positive zero-shot transfer behavior when testing Graph-$Q$-SAT on a task family different from that used for training. While more work is needed to apply Graph-$Q$-SAT to reduce wall clock time in modern SAT solving settings, it is a compelling proof-of-concept showing that RL equipped with Graph Neural Networks can learn a generalizable branching heuristic for SAT search.

## 1 Introduction

Boolean satisfiability (SAT) is an important problem for both industry and academia that impacts various fields, including circuit design, computer security, artificial intelligence and automatic theorem proving. As a result, modern SAT solvers are well crafted, sophisticated, reliable pieces of software that can scale to problems with hundreds of thousands of variables [33].

---

[*]The work was done when the author was a research intern at NVIDIA.

SAT is known to be NP-complete [22], and most state-of-the-art open-source and commercial solvers rely on multiple *heuristics* to speed up the exhaustive search, which is otherwise intractable. These heuristics are usually meticulously crafted using expert domain knowledge and are often iteratively refined via trial and error. In this paper, we investigate how we can use machine learning to improve upon an existing branching heuristic without leveraging domain expertise.

We present Graph-$Q$-SAT, a branching heuristic in a Conflict Driven Clause Learning [40, 21, CDCL] SAT solver trained with value-based reinforcement learning (RL), based on deep $Q$-networks [30, DQN]. Graph-$Q$-SAT uses a graph representation of SAT problems similar to Selsam et al. [39] which provides permutation and variable relabeling invariance. Graph-$Q$-SAT uses a Graph Neural Network [13, 4, GNN] as a function approximator to provide generalization as well as support for a dynamic state-action space. Graph-$Q$-SAT uses a simple state representation and a binary reward that requires no feature engineering or problem domain knowledge. Graph-$Q$-SAT modifies only part of the CDCL based solver, keeping it *complete*, i.e., always yielding a correct solution.

We demonstrate that Graph-$Q$-SAT outperforms Variable State Independent Decaying Sum [31, VSIDS], the most frequently used CDCL branching heuristic, reducing the number of iterations required to solve SAT problems by 2-3X. Graph-$Q$-SAT is trained to examine the structure of the particular problem instance to make better decisions at the beginning of the search, whereas the VSIDS heuristic suffers from poor decisions during the warm-up period.

Our work primarily focuses on the machine learning perspective and thus more work would be required to apply Graph-$Q$-SAT in industrial-scale SAT settings. However, Graph-$Q$-SAT exhibits intriguing properties which might eventually be useful for practical applications. We show that our method generalizes to problems five times larger than those it was trained on. We also show that Graph-$Q$-SAT generalizes across problem types from satisfiable (SAT) to unsatisfiable instances (unSAT). We show that reducing the number of iterations, in turn, could reduce wall clock time, the ultimate goal when designing heuristics. We also show positive zero-shot transfer properties of Graph-$Q$-SAT when the testing task family is different from the training one. Finally, we show that some of these improvements are achieved even when training is limited to a single SAT problem, demonstrating data efficiency.

## 2 Background

### 2.1 SAT problem

A SAT problem involves finding variable assignments such that a propositional logic formula is satisfied or showing that such an assignment does not exist. A propositional formula is a Boolean expression, including Boolean variables, ANDs, ORs and negations. The term literal is used to refer to a variable or its negation. It is convenient to represent Boolean formulas in conjunctive normal form (CNF), i.e., conjunctions (AND) of clauses, where a clause is a disjunction (OR) of literals. An example of a CNF is $(x_1 \vee \neg x_2) \wedge (x_2 \vee \neg x_3)$, where $\wedge, \vee, \neg$ are AND, OR, and negation respectively. This CNF formula has two clauses: $(x_1 \vee \neg x_2)$ and $(x_2 \vee \neg x_3)$. In this work, we use SAT to denote both the Boolean Satisfiability problem and a satisfiable instance, which should be clear from the context. We use unSAT to denote unsatisfiable instances.

There are many types of SAT solvers. We focus on CDCL solvers, MiniSat [10] in particular, because it is an open-source, minimal, but powerful implementation. A CDCL solver repeats the following steps: every iteration it chooses a variable and assigns it a binary value. This is called a decision. Then, the solver simplifies the formula building an implication graph and checks whether a conflict emerged. Given a conflict, the solver can infer (learn) new clauses and backtrack to the variable assignments where the newly learned clause becomes unit (consisting of a single literal). Learnt clauses force a variable assignment which avoids the previous conflict. Sometimes, CDCL solver undoes all the variable assignments keeping the learned clauses to escape futile regions of the search space. This is called a restart.

We focus on the branching heuristic because it is one of the most heavily used during the solution procedure. The branching heuristic is responsible for picking the variable and assigning some value to it. VSIDS [31] is one of the most used CDCL branching heuristics. It is a counter-based heuristic that keeps a scalar value for each literal or variable (MiniSat uses the latter). These values are increased

every time a variable is involved in a conflict. The algorithm behaves greedily with respect to these values called *activities*. Activities are usually initialized with zeroes [27].

## 2.2 Reinforcement Learning

We formulate the RL problem as a Markov decision process (MDP). An MDP is a tuple $\langle \mathcal{S}, \mathcal{A}, \mathcal{R}, \mathcal{T}, \gamma, \rho \rangle$ with a set of states $\mathcal{S}$, a set of actions $\mathcal{A}$, a reward function $\mathcal{R}(s, a, s')$ and the transition function $\mathcal{T}(s, a, s') = p(s, a, s')$, where $p(s, a, s')$ is a probability distribution, $s, s' \in \mathcal{S}, \ a \in \mathcal{A}$. Discount factor $\gamma \in [0, 1)$ weights preferences for immediate reward relative to future reward. The last element of the tuple $\rho$ is the probability distribution over initial states. In the case of *episodic tasks*, the state space is split into the set of non-terminal states and the terminal state $\mathcal{S}^+$. To solve an MDP means to find an optimal policy, a mapping that outputs an action or distribution over actions given a state and which maximizes the expected discounted return $R = \mathbb{E}[\sum_{t=0}^{\infty} \gamma^t r_t]$, where $r_t = \mathcal{R}(s_t, a_t, s_{t+1})$ is the reward for the transition from $s_t$ to $s_{t+1}$. In Section 3 we apply DQN, a value-based RL algorithm that approximates an optimal $Q$-function, an action-value function that estimates the sum of future rewards after taking an action $a$ in state $s$ and following an optimal policy $\pi$ thereafter: $Q^*(s, a) = \mathbb{E}_{\pi, \mathcal{T}, \rho}[\mathcal{R}(s, a, s') + \gamma \max_{a'} Q^*(s', a')]$. A mean squared temporal difference (TD) error is used to make an update step: $L(\theta) = (Q_\theta(s, a) - r - \gamma \max_{a'} Q_{\bar{\theta}}(s', a'))^2$, where $\theta$ parametrizes the $Q$-function. Target network [30] $Q_{\bar{\theta}}$ is used to stabilize DQN. Its weights are copied from the main network $Q_\theta$ after each $k$ minibatch updates.

## 2.3 Graph Neural Networks

Boolean formulas can be of arbitrary size. Moreover, during the solution procedure, some parts of the formula are eliminated and new clauses are added. We need a network architecture which does not assume the input to be of fixed size. Moreover, a Boolean formula should be invariant to the permutation of the clauses, variables and their renaming. To accommodate these requirements and also to take the problem structure into account, we use Graph Neural Networks [13, GNN] to approximate our $Q$-function. We use the formalism of Battaglia et al. [4], which unifies most existing GNN approaches. Under this formalism, GNN is a set of functions that take an annotated graph as input and output a graph with modified annotations but the same topology.

Here, a graph is a directed graph $\langle V, E, U \rangle$, where $V$ is the set of vertices, $E$ is the set of directed edges with $e_{ij} = (i, j) \in E$, $v_i, v_j \in V$, and $U$ is a global attribute which contains the information relevant to the whole graph. We call vertices, edges, and the global attribute entities. Each entity has an associated annotation: $e_{ij} \in \mathbb{R}^e$, $v_i \in \mathbb{R}^v$ or $u \in \mathbb{R}^u$. A GNN changes these annotations as a result of its operations.

A GNN is as a set of six functions: update functions $\phi_e, \phi_v, \phi_u$ and aggregation functions $\rho_{e \to v}, \rho_{e \to u}, \rho_{v \to u}$. The information propagates between vertices along graph edges. Update functions compute new entity annotations. Aggregation functions enable GNN to process graphs of arbitrary topology, compressing multiple entities features into vectors of fixed size. Summation, averaging, taking $\max$ or $\min$ are popular choices of aggregation functions.

More formally, within one iteration, a GNN does the following computations (in order):

$$
\begin{aligned}
\boldsymbol{e}'_{ij} &= \phi_e(\boldsymbol{u}, \boldsymbol{e}, \boldsymbol{v}_i, \boldsymbol{v}_j) \ \forall e_{ij} \in E \\
\boldsymbol{v}'_i &= \phi_v\big[\boldsymbol{u}, \boldsymbol{v}_i, \rho_{e \to v}(\{\boldsymbol{e}_{ki} \mid \forall e_{ki} \in E\})\big] \ \forall v_i \in V \\
\boldsymbol{u}' &= \phi_u\big[\boldsymbol{u}, \rho_{e \to u}(\{\boldsymbol{e}_{ij} \mid \forall e_{ij} \in E\}), \rho_{v \to u}(\{\boldsymbol{v}_i \mid \forall v_i \in V\})\big].
\end{aligned}
$$

A GNN performs multiple iterations to further propagate information in the graph. Neural networks that represent update functions, can be optimised end-to-end using backpropagation.

## 3 Graph-$Q$-SAT

As noted in Section 2.2, we use the MDP formalism for our purposes. Each SAT problem is an MDP sampled from a distribution of SAT problems of a specific family (e.g., random 3-SAT or graph coloring). Moreover, each problem is either satisfiable or unsatisfiable. Hence, a task is defined as follows: $\tau \sim \mathcal{D}(\phi, (un)SAT, n_{vars}, n_{clauses})$, where $\mathcal{D}$ is the distribution of SAT problems with

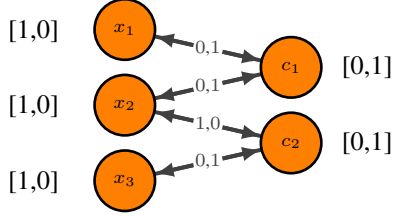
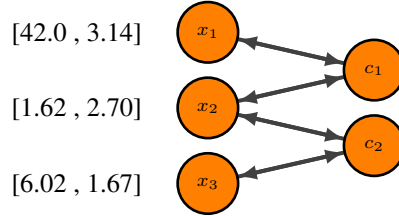

Figure 1: Bipartite graph representation of the Boolean formula $(x_1 \lor x_2) \land (\neg x_2 \lor x_3)$. The numbers next to the vertices distinguish variables and clauses. Edge labels encode literal polarities.

Figure 2: $Q$-function values for setting variables to *true* and *false* respectively. Taking $\arg\max$ across all $Q$-values of variable nodes gives an action.

$\phi$ defining the task family, the second argument defining the problem satisfiability and $n_{vars}$ and $n_{clauses}$ are the number of variables and clauses respectively. Each state of the MDP consists of unassigned variables and unsatisfied clauses containing these variables. The MDP is episodic, and a terminal state is reached when a satisfying assignment is found, or the all possible options have been exhausted, proving unSAT. The action space includes two actions for each unassigned variable: assigning it to true or false. We build upon the MiniSat-based environment of [44]. We modified it to support arbitrary SAT problems and generate a graph representation of the state. It takes the actions, modifies its implication graph internally and returns a new state containing newly learned clauses and without the variables removed during propagation. Strictly speaking, this state is not fully observable. In the case of a conflict, the solver undoes the assignments for variables that are not observed by the agent. However, in practice, this should not inhibit the goal of quickly pruning the search tree: the information in the state is enough to pick a variable that leads to more propagation in the remaining formula. We use a simple reward function: the agent gets a negative reward of $p$ for each non-terminal transition and $0$ for reaching the terminal state. This reward encourages an agent to finish an episode as quickly as possible and does not require elaborate reward shaping.

SAT is an appealing problem from the RL perspective. It has the features that are hard to find in conventional RL environments. First, elements of the state/action set are of different dimensions, which is a challenging case for conventional function approximation techniques. Second, the state-space has a structured, object-oriented representation. Third, SAT allows to vary problem sizes without changing the task family, and to change the task family without changing problem sizes. Lastly, with Hoos and Stützle [18] benchmarks we use, experiments do not take weeks and are easy to iterate on.

### 3.1 State Representation

We represent a SAT problem as a graph similar to Selsam et al. [39]. We make it more compact, using vertices to denote variables instead of literals. We use vertices to encode clauses as well. As Figure 1 shows, our state representation is simple and does not require feature engineering. An edge $(x_i, c_i)$ means that a clause $c_i$ contains literal $x_i$. If a literal contains a negation, a corresponding edge has a $[1, 0]$ label and $[0, 1]$ otherwise. GNNs process directed graphs, so we create two directed edges with the same labels: from a variable to a clause and vice-versa. Vertex features are two-dimensional one-hot vectors, denoting either a variable or a clause. We do not provide any other information to the model. The global attribute input is empty and is only used for message passing.

### 3.2 $Q$-Function Representation

We use the encode-process-decode architecture [4], which we discuss in more detail in Appendix C.1. Similarly to Bapst et al. [3], our GNN labels variable vertices with $Q$-values. Each variable vertex has two actions: set the variable to true or false as shown on Figure 2. We choose the action that gives the maximal $Q$-value across all variable vertices. The graph contains only unassigned variables, so all actions are valid. We use DQN with common techniques such as memory replay, target network, and $\epsilon$-greedy exploration. To expose the agent to more episodes and prevent it from getting stuck, we cap the maximum number of actions per episode similarly to the *episode length* parameter in *gym* [6].

Table 1: Number of MiniSat iterations (no restarts) to solve random 3-SAT instances.

| dataset | median | mean |
|---|---|---|
| SAT 50-218 | 38 | 42 |
| SAT 100-430 | 232 | 286 |
| SAT 250-1065 | 62 192 | 76 120 |
| unSAT 50-218 | 68 | 68 |
| unSAT 100-430 | 587 | 596 |
| unSAT 250-1065 | 178 956 | 182 799 |

Table 2: Graph-$Q$-SAT MRIR trained on SAT-50-218. SAT-50-218 results are for a separate validation set.

| dataset | mean | min | max |
|---|---|---|---|
| SAT 50-218 | 2.46 | 2.26 | 2.72 |
| SAT 100-430 | 3.94 | 3.53 | 4.41 |
| SAT 250-1065 | 3.91 | 2.88 | 5.22 |
| unSAT 50-218 | 2.34 | 2.07 | 2.51 |
| unSAT 100-430 | 2.24 | 1.85 | 2.66 |
| unSAT 250-1065 | 1.54 | 1.30 | 1.64 |

## 3.3 Training and Evaluation

We train our agent using Random 3-SAT instances from the SATLIB benchmark [18]. To measure generalization, we split these data into training, validation, and test sets. To illustrate the problem complexities, Table 1 provides the number of steps it takes MiniSat to solve the problem. Each random 3-SAT problem is denoted as SAT-X-Y or unSAT-X-Y, where SAT means that all problems are satisfiable, unSAT means all problems are unsatisfiable. X and Y stand for the number of variables and clauses in the initial formula. We provide more details about the datasets in Appendix C.2.

While random 3-SAT problems have relatively few variables and clauses, they have an interesting property that makes them more challenging for a solver. For this dataset, the ratio of clauses to variables is close to 4.3 to 1 which is near the *phase transition* at which it is hard to say whether the problem is SAT or unSAT [9]. In 3-SAT problems, each clause has exactly 3 variables. However, learned clauses might be of arbitrary size.

We use Median Relative Iteration Reduction (MRIR) w.r.t. MiniSat as our main performance metric: the number of iterations it takes MiniSat to solve a problem divided by Graph-$Q$-SAT's number of iterations. Similarly to the *median human normalized score* adopted in the Atari domain [16], we use the median instead of the mean to avoid skew from outliers. By one iteration we mean one *decision*, i.e., choosing a variable and setting it to a value. We compare ourselves with the best MiniSat results having run MiniSat with and without restarts. We cap the number of decisions our method takes at the beginning of the solution procedure and then we give control to MiniSat.

We are not interested in the absolute number of iterations per se or the total ratio between VSIDS and Graph-Q-SAT. We use these numbers as a common scale to show the generalisation, transfer and data efficiency properties of our approach.

When training, we evaluate the model every 1000 batch updates on the validation instances and pick the model with the best validation results. After that, we evaluate this model on the test set and report the results. For each model we do 5 training runs and report the average MRIR results, the maximum, and the minimum. We provide all the hyperparameters needed to reproduce our results in Appendix C. Our experimental code as well as the MiniSat *gym* environment can be found at `https://github.com/NVIDIA/GraphQSat`.

## 4 Experimental Results

In this section, we present empirical results for Graph-$Q$-SAT.

## 4.1 Improving upon VSIDS

In our first experiment, we consider whether it is possible to improve upon VSIDS using no domain knowledge, a simple state representation, and a simple reward function. The first row in Table 2 gives a positive answer to that question. DQN equipped with a GNN solves the problems in fewer than half the iterations of MiniSat. Graph-$Q$-SAT makes decisions resulting in more propagations, i.e., inferring variable values based on other variable assignments and clauses. This helps Graph-$Q$-SAT prune the search tree faster. For SAT-50-218, Graph-$Q$-SAT does on average 2.44 more propagations than MiniSat (6.62 versus 4.18). We plot the average number of variable assignments for each problem individually in the Appendix B.

These results raise the question: Why does Graph-$Q$-SAT outperform VSIDS? VSIDS is a counter-based heuristic that takes time to warm up. Our model, on the other hand, perceives the whole problem structure and can make more informed decisions from the beginning. To test this hypothesis, we vary the number of decisions our model makes at the beginning of the solution procedure before we hand the control back to VSIDS. The results in Figure 3 support this hypothesis. Even if our model is used for only the first ten iterations, it still improves performance over VSIDS.

One strength of Graph-$Q$-SAT is that VSIDS keeps being updated while the decisions are made with Graph-$Q$-SAT. We believe that Graph-$Q$-SAT complements VSIDS by providing better quality decisions in the initial phase while VSIDS is warming up. Capping the number of model calls also significantly reduces the main bottleneck of our approach – wall clock time spent on model evaluation.

## 4.2 Generalization Properties of Graph-$Q$-SAT

Next, we consider Graph-$Q$-SAT's generalization properties.

### 4.2.1 Generalization across Problem Sizes

Table 2 shows that Graph-$Q$-SAT has no difficulty generalizing to larger problems, showing almost 4X improvement in iterations for a dataset 5 times bigger than the training set. Graph-$Q$-SAT on average leads to more variable assignments changes per step, e.g., 7.58 vs 5.89 on SAT-100-430 (refer to Appendix B for detailed plots). It might seem surprising that the model performs better for larger problems. However, an increase in score for different problem sizes might also mean that the base solver scales worse than our method does for this benchmark.

### 4.2.2 Generalization from SAT to unSAT

An important characteristic of Graph-$Q$-SAT is that the problem formulation and representation makes it possible to solve unSAT problems when training only on SAT, which is problematic for some existing approaches [39]. The performance is, however, worse than the performance on satisfiable problems. On the one hand, SAT and unSAT problems are different. When the solver finds one satisfying assignment, the problem is solved. For unSAT, the algorithm needs to exhaust all possible options to prove that there is no such assignment. On the other hand, there is one important similarity between the two: an algorithm has to prune the search tree as fast as possible. Our measurements of the average number of propagations per step demonstrate that Graph-$Q$-SAT learns how to prune the tree more efficiently than VSIDS (6.36 vs 4.17 for unSAT-50-218, detailed plots are in Appendix B).

### 4.2.3 Transfer across Task Families

So far, we have examined the generalization properties of Graph-$Q$-SAT varying only the last three arguments of the task distribution defined in Section 3 ($\mathcal{D}(\phi, (un)SAT, n_{vars}, n_{clauses})$). In this section we go one step further and study Graph-$Q$-SAT's *zero-shot transfer* to a new task family $\phi$.

This is a challenging problem. SAT problems have distinct structures, e.g., the graph representation of a random 3-SAT problem looks different than that of a graph coloring problem. GNNs learn graph local properties, i.e. how neighbouring entities' features have a global implication on $Q$-values. It is reasonable to expect a performance drop when changing the task family $\phi$, but the magnitude of the drop gives some indication of the method's ability to transfer across task families. Therefore, we evaluate a model trained on SAT-50-218 on the flat graph coloring benchmark from SATLIB [18]. All the problems in the benchmark are satisfiable. Table 3 shows positive transfer for Graph-$Q$-SAT on the graph coloring benchmark, with MRIR above 1 in five out of eight cases. As expected, MRIR is lower if than in Table 2, where the model was evaluated on the tasks sampled from the same distribution.

Training directly on the graph coloring benchmark indeed improves performance. Graph coloring benchmarks have only 100 problems each, so we do not split them into training/validation/test sets using *flat-75-180* for training and *flat-100-239* to do model selection. Table 4 shows that Graph-$Q$-SAT, trained on flat75-180 shows higher MRIR compared to the transferred model. Additionally, this experiment shows that Graph-$Q$-SAT can scale when training on larger graphs.

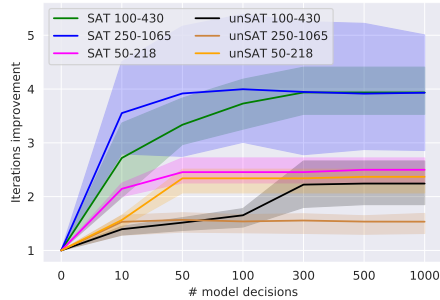
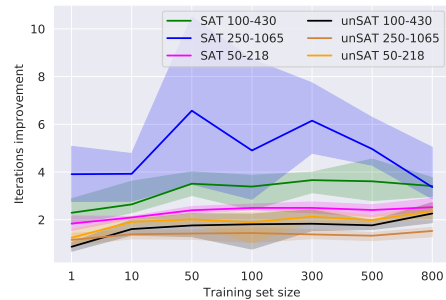

Figure 3: Graph-$Q$-SAT number of maximum first decisions vs performance. Graph-$Q$-SAT shows improvement starting from 10 iterations confirming our hypothesis of VSIDS initialization problem. The shades mark *min* and *max* values.

Figure 4: Dataset size effect on generalization. While Graph-$Q$-SAT profits from more data in most of the cases, it is able to generalize even from one data point. Model is trained on SAT-50-218. The shades mark *min* and *max* values.

Another intriguing property of Graph-$Q$-SAT generalization is that sometimes Graph-$Q$-SAT shows better performance when generalizing in comparison to training from scratch. Learning on SAT-100-430 requires more resources, does not generalize as well, and is generally less stable than training on SAT-50-218 and then transferring to SAT-100-430 and SAT-250-1065. Hence, generalizing with Graph-$Q$-SAT not only reduces the time and samples spent on training, but yields models hardly achievable by learning. We suppose the reason is that transfer is not directly affected by all the issues an RL agent faces when training: higher variance in the returns caused by longer episodes, challenges for temporal credit assignment, and difficulties with exploration.

### 4.3 Data Efficiency

We design our next experiment to understand how many different SAT problems Graph-$Q$-SAT needs to learn from. We varied the SAT-50-218 training set from a single problem to 800 problems. Figure 4 shows that Graph-$Q$-SAT is extremely data efficient. Having more data helps in most cases but, even with a single problem, Graph-$Q$-SAT generalizes across problem sizes and to unSAT instances. This should allow Graph-$Q$-SAT to generalize to new benchmarks without access to many problems from them. We assume that Graph-$Q$-SAT's data efficiency is one of the benefits of using RL. The environment allows the agent to explore diverse regions of state-action space, making it possible to learn useful policies even from a single instance. In supervised learning, data diversity is addressed at the training data generation step.

### 4.4 Wall-Clock Time Bottleneck

The main goal of this work is to show that RL can learn a value function that can be used as a branching heuristic in a SAT solver, and to study the model's generalisation properties. In its current form, more work would be required to apply Graph-$Q$-SAT in an industrial setting, where wall-clock time is the metric of success, and problem sizes are extremely large. However, we believe that Graph-$Q$-SAT should be of interest to the SAT community because reduction of iterations can reduce wall clock time when the number of saved iterations is large enough to tolerate the network inference timings. Due to a shortage of space, we present the wall-clock time and scaling analysis in Appendix A. This analysis shows that MRIR reduction leads to wall clock time improvements on SAT-250 and unSAT-250.

## 5 Related Work

Using machine learning for the SAT problem is not a new idea [14, 15, 12, 42, 47, 28]. Recently, SAT has attracted interest in the deep learning community. There are two main approaches: solving a problem end-to-end or learning heuristics while keeping the algorithm backbone the same. Selsam et al. [39, NeuroSAT] take an end-to-end supervised learning approach demonstrating that GNN can generalize to SAT problems bigger than those used for training. NeuroSAT finds satisfying assign-

Table 3: SAT-50 model's performance on SATLIB flat graph coloring benchmark. The comparison is w.r.t. MiniSat with restarts, since MiniSat performs better in this mode for this benchmark.

| dataset | variables | clauses | Graph-$Q$-SAT MRIR | | |
|---|---|---|---|---|---|
| | | | average | min | max |
| 30-60 | 90 | 300 | 1.51 | 1.25 | 1.65 |
| 50-115 | 150 | 545 | 1.36 | 0.47 | 1.80 |
| 75-180 | 225 | 840 | 1.40 | 0.31 | 2.06 |
| 100-239 | 300 | 1117 | 1.44 | 0.31 | 2.38 |
| 125-301 | 375 | 1403 | 1.02 | 0.32 | 1.87 |
| 150-360 | 450 | 1680 | 0.76 | 0.37 | 1.40 |
| 175-417 | 525 | 1951 | 0.67 | 0.44 | 1.36 |
| 200-479 | 600 | 2237 | 0.67 | 0.54 | 0.87 |

Table 4: Graph-$Q$-SAT MRIR (5 training runs on 75-180, model selection with 100-239).

| dataset | Graph-$Q$-SAT MRIR | | |
|---|---|---|---|
| | average | min | max |
| 75-180 | 2.44 | 2.25 | 2.70 |
| 100-239 | 2.89 | 2.77 | 2.98 |
| 30-60 | 1.74 | 1.33 | 2.00 |
| 50-115 | 2.08 | 2.00 | 2.13 |
| 125-301 | 2.43 | 2.20 | 2.66 |
| 150-360 | 2.07 | 2.00 | 2.11 |
| 175-417 | 1.98 | 1.69 | 2.21 |
| 200-479 | 1.70 | 1.38 | 1.98 |

ments for the SAT formulae and thus cannot generalize from SAT to unSAT problems. Moreover, the method is incomplete and might generate incorrect results, which is extremely important, especially for unSAT problems. Selsam and Bjørner [38] modify NeuroSAT and integrate it into popular SAT solvers to improve timing on SATCOMP-2018 benchmark. While the approach shows its potential to scale to large problems, it requires an extensive training set including over 150,000 data points. Amizadeh et al. [2] propose an end-to-end GNN architecture to solve circuit-SAT problems. While their model never produces false positives, it cannot solve unSAT problems.

The following methods take the second approach: learning a branching heuristic instead of learning an algorithm end-to-end. Jaszczur et al. [19] take the supervised learning approach using the same graph representation as Selsam et al. [39]. The authors show a positive effect of combining DPLL/CDCL solver with the learnt model. As in Selsam et al. [39], their approach requires diligent crafting of the test set. Also, the authors do not compare their approach to the VSIDS heuristic, which is known to be a crucial component of CDCL [23]. Wang and Rompf [44], whose environment we took as a starting point, show that DQN does not generalize for 20-91 3-SAT problems, whereas Alpha(Go) Zero [41] does. Our results show that the issue is related to state representation. They use CNNs, which are not invariant to variable renaming or permutations. Moreover, CNNs require a fixed input size which makes it infeasible when applying to problems with different numbers of variables or clauses.

Yolcu and Póczos [48] use REINFORCE [46] to learn the variable selection heuristic of a local search SAT solver [37]. Their algorithm is an incomplete solver and cannot work with unsatisfiable instances. They also investigate the generalisation over problem sizes on random instances near the phase transition. However, in this experiment, the training problems have ten variables only, and the number of variables in the test set does not exceed 80 with the success ratio of the algorithm staying below the baseline for the latter case.

Lederman et al. [26] train a REINFORCE [46] agent applying GNNs to replace the branching heuristic for Quantified Boolean Formulas (QBF). QBF considers a different problem allowing existential and universal quantifiers. Lederman et al. [26] note positive generalization properties across problem size for problems from similar distributions. Our work focuses more on the generalization and transfer properties of a GNN value-based RL algorithm. We investigate data efficiency properties and merge VSIDS with a trained RL agent, looking into the trade-off between the model use and its effect on the final solution. Apart from that, we show that it is possible to achieve good performance and generalization properties with a simpler state representation. Finally, doing more message propagations per step and using a GNN as a $Q$-function (in their case, a GNN only computes node embeddings) allows us to consider more subtle dependencies in the graph.

Look-ahead SAT solvers [17] perform more computations compared to VSIDS to evaluate the consequences of their decisions. In some of the cases, e.g. random $k$-SAT, this pays off. Difference heuristics used for making a decision measure reduction in the formulae before and after the decision. LRB heuristic [28] uses multi-armed bandits to explicitly optimise for the ability of the variables' to generate learnt clauses. We hypothesise, that Graph-$Q$-SAT might have learnt some aspects of those heuristics (Figure 8 in Appendix B). We believe that integrating Graph-$Q$-SAT with other types of solvers is a promising direction for future research.

Vinyals et al. [43] introduce a recurrent architecture for approximately solving complex problems, such as the Traveling Salesman Problem, approaching it in a supervised way. Bello et al. [5] consider combinatorial optimization problems with RL. Khalil et al. [24] approach combinatorial optimization using GNNs and DQN, learning a heuristic that is later used greedily. It differs from our approach in that their heuristic is effectively the algorithm itself. The environment dynamics in Khalil et al. [24] is straightforward with the next state easily inferred, given the current state and the chosen action. In the case of SAT, there are CDCL steps after the decision, and the next state might be totally different from the current one making the problem harder in terms of learning the $Q$-function. In addition, we use Battaglia et al. [4] which is more expressive than `structure2vec` used in Khalil et al. [24]. The global attribute in Battaglia et al. [4] can facilitate message passing in case of a bigger graph. Having separate updaters for edges and nodes leads to more powerful representations. And, finally, an edge updater of Battaglia et al. [4] can learn better pairwise interaction between the sender and the receiver, enabling sending different messages to different nodes.

Paliwal et al. [34] use GNNs with imitation learning for theorem proving. Carbune et al. [8] propose a general framework of injecting an RL agent into existing algorithms. Cai et al. [7] use RL to find a suboptimal solution that is further refined by another optimization algorithm, in their case, simulated annealing [25, SA]. It is not restricted to SA, and this modularity is valuable. However, it is also a drawback because the second optimization algorithm might benefit more from the first if they were interleaved. For instance, Graph-$Q$-SAT can guide search before VSIDS overcomes its initialization bias.

GNNs have enabled the study of RL agents in state/action spaces of dynamic size, which is crucial for generalization beyond the given task. Wang et al. [45] and Sanchez-Gonzalez et al. [36] consider GNNs for the control problem generalization. Bapst et al. [3] report strong generalization capabilities for the construction task. Multi-agent research [20, 29, 1] shows that GNN benefits from invariance to the number of agents in the team or other environmental entities.

# 6 Conclusions and Future Work

In this paper, we demonstrated that $Q$-learning can be used to learn the branching heuristic of a SAT solver. Graph-$Q$-SAT uses a simple state representation and does not require elaborate reward shaping. We show empirically that Graph-$Q$-SAT causes more variable propagations per step, solving the SAT problem in fewer iterations than VSIDS. For larger problems, we showed that fewer iterations could, in turn, reduce wall-clock time.We demonstrated its generalization abilities, showing more than 2-3X reduction in iterations for problems up to 5X larger and 1.5-2X from SAT to unSAT. We showed how Graph-$Q$-SAT improves VSIDS and that it is data-efficient. We also demonstrated positive transfer properties when changing the task family and showed that training on data from other distributions could lead to further performance improvements.

Although we showed the powerful generalization properties of graph-based RL on SAT, we believe the problem is still far from solved. More work is needed before Graph-$Q$-SAT is ready to compete with branching heuristics in a modern industrial setting. The two main direction of future applied research are scaling and wall-clock time reduction. Some possible ways of tackling these issues include combining the machine learning improvements from above together with an efficient C++ implementation, using a smaller network, reducing the network polling frequency, and replacing the variable activities with Graph-$Q$-SAT's output, similarly to Selsam and Bjørner [38].

From the machine learning perspective, it is intriguing to study how combining benchmarks from different domains might improve the transfer behavior. Further research will focus on scaling Graph-$Q$-SAT using the latest stabilizing techniques [16] and more sophisticated exploration methods. Building an efficient curriculum is another important step towards further scaling the method, motivated by Bapst et al. [3]. Newsham et al. [32] show that the graph structure of SAT problems affects the problem complexity. We are interested in understanding how the structure influences the performance of Graph-$Q$-SAT and how we can exploit this knowledge to improve Graph-$Q$-SAT.

# Broader Impact

We believe that further progress in machine learning can have a profound economic, societal and political impact. It is hard to predict a particular effect of our method on society but, in general, we

believe that the society might benefit from our research through its impact on industry and academia. We consider two examples below.

SAT has a profound impact on circuit design, computer security, artificial intelligence, automatic theorem proving, and combinatorial optimisation, among others. For academia, Graph-$Q$-SAT code and results give a playground to work on GNN scaling, generalisation in RL, transfer and multitask learning, and incorporating a machine learning model with a well established algorithm. It can encourage collaboration between the applied ML and SAT communities. Analysing the behaviour of learned models might give human designers more insights to boost further research.

For industry, having faster SAT solvers would lead to faster production cycles and faster rate of progress as well as to more robust products. In circuitry design, for example, SAT is used for hardware verification. As a result, faster SAT solvers will eventually lead to fewer faults in hardware.

Like any technology, our method also carries potential risks. Further automation might reduce the need for human labour. If not managed and regulated properly, machine learning progress might also exacerbate social and economic inequality.

## Acknowledgments and Disclosure of Funding

The authors would like to thank Rajarshi Roy, Robert Kirby, Yogesh Mahajan, Alex Aiken, Mohammad Shoeybi, Rafael Valle, Sungwon Kim and the rest of the Applied Deep Learning Research team at NVIDIA for useful discussions and feedback. The authors would also like to thank Andrew Tao and Guy Peled for providing computing support. The authors thank Henry Kenlay for useful discussions on GNN scaling properties.

Vitaly Kurin is a doctoral student at the University of Oxford funded by Samsung R&D Institute UK through the *Autonomous Intelligent Machines and Systems* program. This work was done during Vitaly's internship at NVIDIA. Saad Godil is a Director of Applied Deep Learning Research at NVIDIA. Shimon Whiteson is a Professor of Computer science at the University of Oxford and the Head of Research at Waymo UK. Shimon Whiteson has received funding from the European Research Council under the European Union's Horizon 2020 research and innovation programme (grant agreement number 637713). Bryan Catanzaro is a Vice President, Applied Deep Learning Research at NVIDIA.

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
