[Supplementary Material]

# A   Wall-clock Time and Scaling Analysis

(a) SAT-250

(b) unSAT-250

Figure 5: Graph-$Q$-SAT's MRIR improvement (10 model calls) results in the wall clock time reduction. The curves show the averaged performance across five runs with the shade denoting the worst and the best runs.

Figure 5 demonstrates that reduction in the number of iterations together with limiting the number of model calls results in wall clock time improvements for the datasets where the number of saved iterations is large enough to tolerate the network inference timings.

More generally, we can anticipate the settings in which Graph-$Q$-SAT will yield an improvement in wall clock time over VSIDS by analyzing the factors contributing to their runtime performance. Assuming VSIDS's computational cost is negligible, we can compute the wall clock time of MiniSat with VSIDS as follows: $W_{VSIDS} = \sum_{t=1}^{T} P(t)$, where $P(t)$ is the unit propagation time (a procedure of formula simplification after the branching decision is made). Similarly, Graph-$Q$-SAT saves some fraction of iterations at the cost of added neural network inference time: $W_{\text{Graph-}Q\text{-SAT}} = \sum_{t=1}^{T/S} P(t) + \sum_{t=1}^{K} I(t)$, where $S$ is the reduction of the number of iterations, $K$ is the number of our model forward passes ($K << T$ for larger problems), and $I(t)$ is the network inference time. Thus, Graph-$Q$-SAT leads to wall clock speed ups when the total inference time stays below the time spent on propagation for the reduced number of VSIDS decisions. This seems plausible assuming that $T$'s growth is unbounded, $K << T$ and linear dependence of $I(t)$ on the number of vertices. To check the linear dependence, we generated $10^5$ graphs with characteristics similar to random 3-SAT problems (bipartite graph, each variable is connected to 13 clauses, and clause/variable ratio is 4). Figure 6 confirms that the dependence is linear.

# B   Propagations per step

Figure 8 shows that on average using Graph-$Q$-SAT leads to more propagations per step than VSIDS.

# C   Reproducibility

We implement our models using Pytorch [3] and Pytorch Geometric [2].

## C.1   Model architecture

We use Encoder-Process-Decode architecture from [1]. Encoder and decoder are independent graph networks, i.e. MLPs taking whole vertex or edge feature matrix as a batch without message passing. We call the middle part 'the core'. The output of the core is concatenated with the output of the encoder and gets fed to the core again. We describe all the hyperparameters in Appendix C.3. We also plan to release the experimental code and the modified version of MiniSat to use as a gym environment.

Figure 6: Graph-$Q$-SAT inference time linearly depends on the number of vertices in the graph.

Figure 7: Encode-Process-Decode architecture. Encoder and Decoder are independent graph networks, i.e. MLPs taking whole vertex/edge data array as a batch. $k$ is the index of a message passing iteration. When concatenating for the first time, encoder output is concatenated with zeros.

## C.2 Dataset

We split SAT-50-218 into three subsets: 800 training problems, 100 validation and 100 test problems. For generalization experiments, we use 100 problems from all the other benchmarks.

For graph colouring experiments, we train our models using all problems from flat-75-180 dataset. We select a model, given the performance on all 100 problems from flat-100-239. So, evaluation on these two datasets should not be used to judge the performance of the method, and they are shown separately in Table 4. All the data from the second part of the table was not seen by the model during training (flat-30-60, flat-50-115, flat-125-301, flat-150-360, flat-175-417, flat-200-479).

## C.3 Hyperparameters

Table 5 contains all the hyperparameters necessary to replicate our results.

## C.4 Graph-$Q$-SAT pseudocode

(a) SAT 50-218

(b) unSAT 50-218

(c) SAT 100-430

(d) SAT 100-430

Figure 8: Average number of variable assignments change per step for (un)SAT-50-218 and (un)SAT-100-430.

---

**Algorithm 1** Graph-$Q$-SAT Action Selection

---

**Input:** graph network $GN_\theta$, state graph $G_s := (V, E, U)$,
with vertex features $V = [V_{vars}, V_{clauses}]$, edge features $E$, and a global compontent $U$.

---

$V', E', U' = GN(V, E, U)$;
$VarIndex, VarPolarity = \arg\max_{ij} V'_{vars}$;
**Return** $VarIndex, VarPolarity$;

Table 5: Graph-$Q$-SAT hyperparameters.

| Hyperparameter | Value | Comment |
|---|---|---|
| *DQN* | | |
| – Batch updates | 50 000 | |
| – Learning rate | 0.00002 | |
| – Batch size | 64 | |
| – Memory replay size | 20 000 | |
| – Initial exploration $\epsilon$ | 1.0 | |
| – Final exploration $\epsilon$ | 0.01 | |
| – Exploration decay | 30 000 | Environment steps. |
| – Initial exploration steps | 5000 | Environment steps, filling the buffer, no training. |
| – Discounting $\gamma$ | 0.99 | |
| – Update frequency | 4 | Every 4th environment step. |
| – Target update frequency | 10 | |
| – Max decisions allowed for training | 500 | Used a safety against being stuck at the episode. |
| – Max decisions allowed for testing | 500 | Varied among [0, 10, 50, 100, 300, 500, 1000] for the experiment on Figure 3. |
| – Step penalty size $p$ | -0.1 | |
| *Optimization* | | |
| – Optimizer | Adam | |
| – Adam betas | 0.9, 0.999 | Pytorch default. |
| – Adam eps | 1e-08 | Pytorch default. |
| – Gradient clipping | 1.0 | 0.1 for training on the graph coloring dataset. |
| – Gradient clipping norm | $L_2$ | |
| – Evaluation frequency | 1000 | |
| *Graph Network* | | |
| – Message passing iterations | 4 | |
| – Number of hidden layers for GN core | 1 | |
| – Number of units in GN core | 64 | |
| – Encoder output dimensions | 32 | For vertex, edge and global updater. |
| – Core output dimensions | 64,64,32 | For vertex, edge and global respectively. |
| – Decoder output dimensions | 32 | For vertex updater, since only Q values are used, no need for edge/global updater. |
| – Activation function | ReLU | For everything but the output transformation. |
| – Edge to vertex aggregator $\rho_{e \rightarrow v}$ | sum | |
| – Variable to global aggregator $\rho_{v \rightarrow u}$ | average | |
| – Edge to global aggregator $\rho_{e \rightarrow u}$ | average | |
| – Normalization | Layer Normalization | After each GN updater |

---
**Algorithm 2** Graph-$Q$-SAT Training Procedure
---
**Input:** Set of tasks $\mathcal{S} \sim \mathcal{D}(\phi, (un)SAT, n_{vars}, n_{clauses})$ split into $\{\mathcal{S}_{train}, \mathcal{S}_{validation}, \mathcal{S}_{test}\}$, $\phi$ is the task family (e.g. random 3-SAT, graph coloring). All hyperparameters are from Table 5.
Randomly Initialize Q-network $GN_\theta$;
$updates = 0$;
$totalEnvSteps = 0$;
**repeat**
  **repeat**
    Sample a SAT problem $p \sim \mathcal{S}_{train}$;
    Initialize the environment $env = SatEnv(p)$;
    Reset the environment $s = env.reset()$;
    take action
    $a = \begin{cases} random(\mathcal{A}), \text{with probability } \epsilon \\ selectAction(s), \text{with probability } 1 - \epsilon \end{cases}$
    Take env step $s', r, done = env.step(a)$;
    $totalEnvSteps + = 1$;
    dump experience $buffer.add(s, s', r, done, a)$;
    **if** $totalEnvSteps$ mod $updateFreq == 0$; **then**
      Do a DQN update;
    **end if**
    **if** $totalEnvSteps$ mod $validateFreq == 0$; **then**
      Evaluate $GN_\theta$ on $\mathcal{S}_{validation}$;
    **end if**
  **until** Proved SAT/unSAT ($done$ is *True*)
**until** $updates == totalBatchUpdates$
Pick the best model $GN_\theta$ given validation scores;
Test the model $GN_\theta$ on $\mathcal{S}_{test}$;
---