[Reviews · NeurIPS 2020]

Review 1

Summary and Contributions: The paper proposes a branching heuristic, Graph-Q-SAT, for a complete Conflict Driven Clause Learning (CDCL) SAT solver. Graph-Q-SAT is learned using RL combined with the Sat Solver MiniSat via deep Q-networks, where a graph neural network (GNN) is used to approximate the Q function. The authors experimentally showed that Graph-Q-SAT outperforms the popular Variable State Independent Decaying Sum (VSIDS) heuristic, i.e., Graph-Q-SAT requires fewer number of iterations to solve SAT problems when it was trained within the same domain. They also showed some positive results in cross-domain transfer learning for graph coloring.

Strengths: The paper experimentally shows as a proof of concept that deep Q-networks are able to learn a useful heuristic for SAT solvers. They show that there is potential in learning a better heuristic branching function for SAT Solver minisat. They focus mainly on the increased unit propagation that the learned heuristic provides. The approach is fairly variable and clause independent and can be trained on one size formula and applied to different size instances and even benefit in some other domains. It mainly helps early on in the search (top of the search tree) because it looks at the variables more globally than the standard heuristic they compare to, the VSIDS heuristic in minisat. It is interesting to see that the learned heuristic is able to generalize a bit across problem sizes, from SAT to unSAT, and to graph coloring. In particular it is interesting to see that the framework is applicable also to UNSAT.

Weaknesses: There have been several “proof-of-concept” papers using deep learning for SAT. This paper is an interesting paper, but yet another proof of concept, using relatively small size SAT instances. The paper falls short in terms of showing a true potential for improving the state of the art of Sat Solvers. The experimental section is limited: they mainly consider random 3SAT instances (at the phase transition, which is good) but they are relatively small instances (up to 250 variables when SAT solvers can solve considerably larger problems for satisfiable instances with thousands of variables and millions of clauses and hundreds of variables and thousands of clauses for unsat (see e.g., 2016 SAT competition)). They also consider graph coloring instances but again not very large size problems. I would like to see a broader class of benchmark problems in terms of sizes but also domains, in particular more structured instances. Also, it is too early to claim that the proposed method is data-efficient, since only a 3-SAT benchmark is tested for this without addressing scalability issues properly. The paper focuses on a single heuristic (VSIDS) and a single solver (MINISAT). What about other SAT solvers and heuristics? In addition, the paper does not provide a good perspective on the scalability of the method and tradeoffs between inference and search. I'm most skeptical about the wall-clock time analysis. This seems to be an extrapolation argument that hinges on several assumptions. A key design choice for the best current SAT solvers is to keep the heuristic as simple as possible so that the solver can do as many variable assignments and unit props as possible per second. It will be hard to create an overall more efficient solver with any learned model that is significantly slower to evaluate than the current fast heuristics. Again, better heuristics in terms of pruning are known but not used because they are too slow to evaluate. So, it remains unclear whether this approach is the path to a better practical SAT solver but the paper does give some evidence that there is hope. It would have been more convincing if the method was shown on more structured problems. That challenge may be quite different. For random style problems, a uniform heuristic may be easier to learn. So this an ok proof of concept but says little about how to use it effectively and how it scales up to more challenging SAT instances and real word instances.

Correctness: The method looks correct to me, but I have doubt about the claims that the proposed method is data efficient. Additional experiments are required for such claims. See also the above comments.

Clarity: The paper is generally well written. However, the paper can be improved in the description of the algorithms – while one can understand at a high level the approach, it is not easy to follow what the model is doing after reading the main paper since the information is scattered in the main paper and SI. I suggest the authors provide in the main paper high level pseudocode about the overall approach and integration with Minisat with understandable description of he function calls . Even the SI is not self-contained, it relies too much on pointing to other papers and just providing high level diagrams. The SI should provide better explanations and provide more details including figure 7 and also more detailed pseudocode of Algorithm2 with an explanation of the function calls. You have room in the Si for that.

Relation to Prior Work: The relation of the proposed framework to prior work is overall well described. They cited paper [22] as related to the framework proposed here. They say: “It differs from our approach in that their heuristic is effectively the algorithm itself.” Indeed the approach in this paper seems to be similar to the one used in papee 22. Adapting the deep Q networks in [22] to learn heuristic for CDCL looks straightforward to me. So the authors should provide a deeper discussion concerning the novel aspects of their approach.

Reproducibility: Yes

Additional Feedback: We encourage the authors to consider different categories of SAT instances: it is likely that you can improve performance by learning across categories. Also, random kSAT instances are probably not the best category from a learning perspective since they lack the structure and there is a theoretical lower bound on the size of the resolution proof (an exponential lower bound on the size of the search tree). So not clear how much the heuristics can be improved for random instances. Real-world instances, on the other hand, have a much richer structure, they often are characterized by small backdoors (and therefore restarts are very effective) so it’s more likely to see more impressive gains from a learning methodology. Additional thoughts: A key reason why modern SAT solvers use simple branching heuristics is because they want to set variables as fast as possible. The VSIDS heuristic is pretty good for that. But, for random SAT (and related random graph coloring problems) VSIDS is not considered to be a good heuristic. So using VSIDS as a baseline method for random instances is not a good comparison. In fact, we can see from the experiments in [1], the benefits from VSIDS for small, random instances are rather insignificant. One of the fastest solvers for those instances from a while back is the SATZ solver (not a CDCL solver) and this solver, as part of the heuristic choice, actually probed variable settings to look for the most propagation. This is expensive but works well for various kinds of random style problem instances, where clause learning is not very useful. So, SATZ can go up to 500/600 or more variables for hard random UNSAT instances. I suspect the Graph-Q-SAT is learning some aspects of that heuristic. It's still interesting that this can be learned but does not yet push us into new territory. At least the paper does not show that properly given the small-scale experiments. Another example: In the earlier days of SAT research, Operations Research people thought they would crush the AI researchers by using sophisticated “pruning” techniques based on sophisticated Linear Programming (LP) relaxations, but soon they realized that the results were quite poor given that the LP-based pruning was not much better than what one would get with unit propagation; but while unit propagation is very fast, running LP is much more expensive – so in the end the expensive inference based n LP was not worth it. [2] is another paper that uses deep learning to facilitate SAT solving and they perform experiments on industrial instances. So in summary this work is promising. Post-Rebuttal ========== Thank you for the rebuttal. Indeed, I understand and agree to some extent with the authors that this is a proof of concept. Frankly I’d like to see more papers like this combining RL and deep learning for SAT at NeurIPS, and this paper does introduce some interesting ideas and results, in particular for UNSAT instances. So, I’m increasing my score to 7 and I support acceptance of the paper.


Review 2

Summary and Contributions: This paper presents Graph-Q-SAT, a system that learns to guide branching decisions in a complete SAT solver (namely, MiniSAT). The authors cast solving a SAT instance as an MDP, where states are defined by partial variable assignments and an action involves setting another variable. A graph neural network, that accepts the graph structure of the SAT formula being solved as input, is used as a function approximator (the graph reflects simplifications made as variables are set, as well as learned clauses that are added, as the solver progresses). After training on a set of formulas, the learned DQN agent is used to guide early branching decisions in MiniSAT -- in situations before the traditional VSIDS heuristic has had a chance to "warm up". The authors demonstrate that their system is able to boost the performance of the underlying solver, as measured by the median reduction in the number of branching decisions. They also demonstrate their system's success on a number of transfer tasks: from smaller SAT formulas to larger ones, from SAT formulas to UNSAT formulas, and to some extent, from formulas drawn from one problem family to another.

Strengths: This paper contributes to a growing body of research at the intersection of ML and combinatorial optimization. Many such papers have appeared at recent NeurIPS conferences, and I expect that there will be broad interest in the results presented here. It is noteworthy that the authors demonstrate a significant improvement over the VSIDS heuristic built into MiniSAT -- this has been the state-of-the-art heuristic for decades in complete SAT solvers, and improving on it is a notable accomplishment. Further, the fact that the gains come in random 3SAT instances, a traditional Achilles Heel for complete solvers (which generally tend to perform better in instances with structure, like those arising in industrial domains) is also interesting. The relative simplicity of the approach is also a plus: the authors use standard tools from the ML toolkit -- Q-learning and GNNs. The state representation is intuitive and the reward structure is straightforward. The novelty in the paper arises primarily from the application domain (specifically, the problem formulation) and the quality of the execution. The empirical results, as presented, are convincing -- though, as the authors themselves admit, this is still a proof-of-concept, and much work remains to be done before ML augmented solvers are competitive with traditional solvers when performance is measured using wall-clock time.

Weaknesses: One of the weaknesses of this paper is one that is already acknowledged by the authors -- the fact that the gains in the number of branching decisions from using Graph-Q-SAT do not translate into faster solution times (as measured by wallclock time). However, given the exploratory nature of this work, I'm willing to look past this issue. A bigger issue lies with some of the technology used: MiniSAT and SATLIB problem instances. The SATLIB library at this point is nearly 20 years old, and there are much more recent problem compilations that the authors could have used. Similarly, while MiniSAT is a well-established solver in the community, it has been surpassed by more modern replacements such as Glucose, which might have been a better candidate for a baseline solver. I'm not sure these issues are fatal, however, and I'm willing to give the authors the benefit of the doubt, as the combination of ideas presented here is still worthy of wider dissemination.

Correctness: Yes, and the authors acknowledge the limitations of their empirical results, which is much appreciated.

Clarity: I found the paper very well-written and easy to follow. Some minor typos: Line 58: "'x' or 'NOT x' makes up a literal" --> "The term literal is used to refer to a variable or its negation." Lines 61--62: "This CNF has" --> "This CNF formula has" Line 67: "it chooses a literal and assigns a variable a binary value" --> "it chooses a variable and assigns it a truth value" Lines 69--72: The sentence on these lines is a little difficult to parse; I'd advise rephrasing or breaking it up into multiple sentences. Line 348: "incorporating a machine learning model to" --> "incorporating a machine learning model with"

Relation to Prior Work: Yes, and it's very thorough.

Reproducibility: Yes

Additional Feedback: While the current set of experiments fails to establish Graph-Q-SAT's superiority over MiniSAT on wallclock time, there is one potential work-around: the authors could investigate whether Graph-Q-SAT has a *qualitative* advantage, i.e., given the ML boost, are there SAT instances that Graph-Q-SAT can solve that MiniSAT cannot? For example, larger random 3SAT instances near the phase transition? Or UNSAT instances? If so, then such results could make this paper stronger.


Review 3

Summary and Contributions: This paper seeks to investigate whether RL combined with GNN can lead to better branching heuristics for CDCL based SAT solvers. The key metric that authors focus on is the number of decisions, and authors demonstrate that the number of decisions taken by CDCL solver reduces when using the decisions based on Graph-Q-SAT in the early part of the search when the VSIDS is "warming up".

Strengths: The paper seeks to address one of the most fundamental challenge in modern SAT solving: design of efficient branching heuristics. The paper also identifies that one can improve the initial warm up phase where VSIDS scores are more or less meaningless.

Weaknesses: There are several weaknesses in the approach that need to be addressed: 1. The power of VSIDS is not in reducing the number of decisions. In fact, look-ahead based heuristics are known to be efficient in reducing the number of decisions but in modern CDCL solvers, its the combination of complexity of the branching heuristics and its impact. VSIDS is preferred because it is very easy to maintain, it does not require one to do a global look. Another issue is that CDCL solvers are not well equipped to work for random SAT instances. Local search techniques perform much better and therefore, drawing any inference from random SAT instances is typically a dangerous proposition. It is typical in SAT community that a new heuristics may improve on one or two class of benchmarks but it is not generally robust. I would strongly suggest that authors either focus on SAT Race/SAT competition benchmarks. If authors want to focus on random instances, it is perhaps more instructive to compare with survey propagation and local search techniques.

Correctness: While the general approach of attempting to improve on branching heuristics is a good research direction (one that consistently sees new papers in SAT conference and certainly has potential to lead to breakthroughs), the methodology of this paper needs substantial revision before claims can be evaluated.

Clarity: The paper is well written and I was able to understand the core content of the paper

Relation to Prior Work: It would be perhaps unfair to ask authors to refer to long line of work in SAT community but at the same time it is important that authors refer to look ahead solving techniques in regards to decision heuristics, some of the recent improvements such as LRB based heuristics.

Reproducibility: Yes

Additional Feedback: Thank you for the rebuttal. The rebuttal raises very valid issue of how to judge this paper in the context of NeurIPS being a ML venue. That said, the reviewer does not have expertise on this matter. From the perspective of SAT, the paper has significant shortcoming that need to be overcome in terms of comparisons based on either non-random benchmarks or comparisons with respect to local search techniques. I have increased the score from 3 to 4 to reflect my opinion that AC/SAC are probably in a better position to make decision on how to handle this paper from the perspective of an ML conference.

[Author Response · NeurIPS 2020]

We thank the reviewers for their constructive feedback. However, we strongly believe that our work has been misinterpreted by the reviewers. We do not claim that we solved the boolean SAT problem, as we stress in the paper. If we did, we would have submitted to an applied SAT conference. We chose NeurIPS because, as we say in the introduction, we are mainly interested in the machine learning (ML) side of the problem. And we believe that our work should be judged based on its two key contributions.

First, our paper is not a statement that Graph-Q-SAT is a state of the art SAT heuristic. It is a proof of concept, and we should not expect state-of-the-art performance right out of the gate. From this perspective, we believe that our paper should be published because it provides a proof-of-concept validation for a very different approach to devising heuristics for SAT solvers. This validation was not trivial to obtain and opens up a lot of new research opportunities, some of which ultimately could improve the state of the art on SAT. But that won't happen if the paper is not published.

Second, our contribution from the ML perspective is answering the question in the title. It is the study of the properties of a value-based RL algorithm arising in a challenging setting of the boolean SAT problem, studying its generalisation and transfer behaviour, and its data efficiency properties. From this perspective, SAT is not more than a gym environment, similar to how the researchers in the field use MuJoCo or Atari – a simplification enabling the study of some abstract problems. We use boolean SAT because it has the features that are hard to find in conventional RL environments:

- Elements of the state/action set are of different dimensions, which is a challenging case for conventional function approximation techniques.
- Structured, object-oriented state-space representation.
- Ability to vary problem sizes without changing the task family, and to change the task family without changing problem sizes.
- Experiments do not take weeks and are easy to iterate on.

Here, we are not interested in the absolute number of iterations per se or the total ratio between VSIDS and Graph-Q-SAT. We use these numbers as a common scale to show the generalisation, transfer and data efficiency properties of our approach. We believe that our contributions are significant for NeurIPS community because:

- we show surprising generalisation properties of a deep RL algorithm that are usually considered brittle and overfitting for the training task;
- we show the data efficiency properties of the algorithm, which pinpoints the attractiveness of RL compared to the supervised learning approach;
- we demonstrate positive transfer behaviour, which is a rare thing in deep RL. Often, the task differences studied in transfer for RL are as small as different reward functions. In case of structured state/action spaces, adding two legs to an agent and showing the score higher than of a random policy is considered an achievement.
- the code of the environment we improve and refactor can spark further development of the algorithms in multi-task RL, curriculum, transfer learning and other subfields of RL dealing with multiple tasks.

We submit that our paper deserves a much higher score at an ML conference and we think that our experiments are sufficiently thorough and easy to replicate.

**Response to specific comments**    In this section, we address some specific comments from the reviewers. We will address the rest in the final version of our paper.

*"Adapting the deep Q networks in [22] to learn heuristic for CDCL looks straightforward to me. So the authors should provide a deeper discussion concerning the novel aspects of their approach."*

The problem studied in [22] is much easier from the RL perspective than the one considered in our work. The environment dynamics in [22] is straightforward with the next state easily inferred, given the current state and the chosen action. In our case, there are CDCL steps after the decision, and the next state might be totally different from the current one making the problem harder in terms of learning the Q-function. We disagree that adapting DQN to our case is a straightforward task, and our work has to address two nontrivial issues. First, most of the existing insights about DQN are related to its main testbed — Atari 2600. And many pieces of the common knowledge did not hold in our case. For example, the target network, which is considered of utmost importance for DQN, was harmful to us when updated less frequently. The same holds about the memory replay size. We found out that large replay buffer size is destructive in our setting, which was not discussed before in the literature. Second, GNN research, especially using them in deep RL, is a young area. In our work, we use [4], which is more expressive than `structure2vec` used in [22]. The global attribute in [4] can facilitate message passing in case of a bigger graph. Having separate updaters for edges and nodes leads to more powerful representations. And, finally, an edge updater in [4] can learn better pairwise interaction between the sender and the receiver, enabling sending different messages to different nodes.

[Meta-Review · NeurIPS 2020]

The paper describes a new branching heuristic based on GNNs with DQNs. This is novel and promising. From a SAT perspective, the approach is not compared to the state of the art, but it provides a useful proof of concept that illustrates how GNNs and DQNs can be used to reduce the number of iterations of the branching heuristic. Due to the high computational cost of GNNs and DQNs this does not translate into a reduction in computation time, but the ideas are still useful and promising.